# How does digital village construction influences carbon emission? The case of China

**Aimin Hao, Yirui Hou◉\*, Jiayin Tan**

School of Economics, Zhengzhou University of Aeronautics, Zhengzhou, Henan, China

\* ll17839986737@163.com

## Abstract

Taking 30 provinces in China from 2011 to 2020 as a research sample, this paper empirically tests the impact of digital village construction on carbon emissions. This study found that there is an "inverted U" curve relationship between digital rural construction and rural carbon emissions. Agricultural planting structure and agricultural technology efficiency are important ways for digital village construction to reduce agricultural carbon emissions. The study also found that the higher the level of economic development, the stronger the carbon emission reduction effect of digital village construction. In addition, there are also significant differences in the carbon emission reduction effect of digital village construction in regions with different environmental regulation intensities. Finally, in terms of the relationship between digital economic activities and carbon emission reduction, the research conclusions of this paper have important implications.

**Data Availability Statement:** All relevant data are within the paper and its Supporting Information file.

## Introduction

The world has entered the era of global climate change, which has become the biggest non-traditional security challenge facing human development. At present, the global problem that countries are concerned about is the massive emission of carbon dioxide ($CO_2$) and other greenhouse gases. With the deepening understanding of the relationship between greenhouse gas emissions and climate change, the call for the international community to take countermeasures to reduce emissions is getting louder.

China is the world's most populous country and the world's largest carbon emitter. Coping with carbon emission reduction is the biggest challenge for China and the world to achieve sustainable development. Therefore, it is undoubtedly of great significance to explore an effective path for China to achieve the goal of carbon neutrality.

At present, the main challenge facing China is that the time from carbon peak to carbon neutrality is "too short". As of 2021, 132 countries and regions around the world have proposed the time to achieve carbon neutrality. In terms of timing, most countries propose to achieve carbon neutrality by 2050. In terms of period, it takes an average of more than 50 years for countries to reach carbon neutrality from the carbon peak.

**Funding:** The Graduate Innovation Fund Project of Zhengzhou University of Aeronautics, 2022CX23, Yirui Hou. The National Social Science Foundation of China, 19BJY130, Aimin Hao.

**Competing interests:** The authors have declared that no competing interests exist.

China has announced that the interval from carbon peak to carbon neutrality is 30 years, however, the EU promised a time of 60 to 70 years. The interval time of the latter is more than 2 times that of the former. This indicates that China needs to promote energy conservation and emission reduction targets in various industries and regions in an orderly manner with faster speed and higher efficiency.

At present, what needs to be explored is whether China's digital village construction can affect the carbon neutrality and carbon peaking strategy, and if so, what are the characteristics of the economic mechanism and effect behind it? In China's specific context, does the relationship between digital village construction and rural carbon emissions also follow the environmental Kuznets curve (EKC) hypothesis? The exploration of these issues is not only related to the realization of China's "dual carbon" goal strategy but also has important reference significance for all countries in the world to achieve the goal of carbon neutrality.

From the existing research, most of the influencing factors of energy-saving and emission reduction take cities as the research objects and pay less attention to agriculture and village. However, village energy issues are increasingly becoming an important source of increased regional carbon emissions. According to the China Energy Statistical Yearbook, village energy consumption has more than doubled in the past two decades. There are prominent problems in village areas, such as backward infrastructure, relatively lagging energy socialization services, unreasonable energy consumption structure, and low degree of waste utilization. These problems seriously hinder the realization of energy conservation, emission reduction, and sustainable development goals. It is worth noting that with the development of internet technology, digital village construction based on digital elements has gradually demonstrated its carbon emission reduction potential. However, according to the literature available to the author, few studies have paid attention to the impact of digital village construction on carbon emissions.

Compared with the existing research, the possible marginal contributions of this paper are as follows: First, this paper introduces the digital economy as technological advancement into the Solow growth model and proposes the theoretical hypothesis that there is an "inverted U" curve relationship between digital villages construction and carbon emissions. Secondly, this paper quantitatively measures China's rural carbon emissions from three perspectives: production, life, and ecology. Based on the regional economic development level and the intensity of environmental regulation, the heterogeneity test was carried out. The research in this paper enriches the empirical evidence on the impact of digital economic activities on carbon emissions.

The rest of this paper is organized as follows. The second section presents a review of the literature on the digital economy and carbon emissions. The third section provides the theoretical analysis. The fourth section describes the empirical methods and data resources. The fifth section presents the empirical results and discussions. The last section offers the conclusion and implications.

## Literature review

### Research on environmental effects of economic activities

Existing studies have paid less attention to carbon emission reductions from rural economic activities. The related literature can be divided into two categories. The first is the relationship between economic growth and environmental quality. This involves the famous Kuznets theory [1]. The core of this series of literary studies is to incorporate environmental quality as an output target for economic growth. Another line of literature is to study the effects of environmental externalities arising from different economic activities. Factors such as urbanization

[2], FDI [3], and economic agglomeration [4] are the focus of scholars' consideration. At the same time, with the development of information technology and the popularization of the Internet, the impact of digital economic activities on the environment and sustainable development has attracted more and more attention from scholars.

There is no consensus on the impact of digital economic activity on carbon emissions. On the one hand, some scholars believe that the digital economy can achieve the goal of reducing carbon emissions by improving the energy structure [5]. Moreover, the carbon emission reduction effect of digital economic activities shows heterogeneity with the change in the economic circle, and there is a spatial spillover effect [6]. The intervention of digital elements can not only promote the transformation of industrial development to green and low-carbon but also effectively solve technical problems such as emission monitoring [7]. The application of Internet communication technology is good for environmental improvement [8]. As an important platform for disseminating information on pollution prevention and control, the Internet often uses informal channels to improve air quality [9]. However, on the other hand, some scholars have proposed that the use of digital technology will increase power consumption [10]. Especially in regions where the energy structure is biased toward fossil fuels, the total carbon emissions have increased significantly [11].

## Research on digital village construction

China's "Digital Village Development Strategy Outline" pointed out that "Digital village construction is the application of digital technology in agricultural and rural economic and social development. As well as the endogenous agricultural and rural modernization development and transformation process due to the improvement of farmers' modern information skills." As a key path to reshaping rural economic and social development, digital village construction is an important driving force for promoting the transformation of low-carbon and green development in agriculture and rural areas. The current research on digital village construction mainly focuses on measurement.

There is no consensus on the measurement method. Zeng et al. [12] constructed an evaluation index system for digital village construction from five perspectives: digital infrastructure, data resources, digital industrialization, industrial digitization, and governance digitization. Qi et al. [13] used Internet penetration, the development level of rural e-commerce, and the development of rural inclusive finance as indicators to measure the construction of digital villages. Cui and Feng [14] used digital environment, digital investment, digital benefit, and digital service as evaluation indicators to construct a digital rural economic index system. Studies have shown that digital village construction is conducive to optimizing the efficiency of resource allocation and improving traditional agricultural production methods [15]. Digital village construction is increasingly becoming an important engine for the transformation of agricultural and rural development.

Through literature review, it is found that the existing literature affirms the importance of digital elements for the green and low-carbon transformation of economic development. However, few scholars have studied the carbon emission reduction effect of digital village construction from the perspective of the digital and low-carbon transformation of agriculture and rural areas. Logically, the various positive externalities generated by the application of digital technology in agriculture have an important impact on rural economic growth, energy use efficiency, and pollutant emissions. Examining the energy-saving and emission-reduction effects of digital village construction is conducive to a more comprehensive understanding of the green economic effects of digital production factors. Therefore, on the basis of theoretical analysis, this paper uses Chinese provincial data to empirically test the impact of digital village

construction on carbon emissions and its mechanism, in order to provide experience and reference for achieving the goal of carbon neutrality.

## Theoretical analysis

### Typical facts about the impact of digital village construction on carbon emissions

First, digital village construction provides a basic guarantee for the low-carbon development of agriculture. Based on technological innovation, it deeply integrates digital technologies such as big data, AI, and cloud computing with the entire agricultural industry chain and ecological protection. This will help to form a high-quality and stable rural industrial green ecosystem. The typical fact is that the traditional agricultural production mode of "depending on the weather" is transformed into a "controllable" intelligent production mode. Through precision planting and breeding, the carbon emissions of the agricultural system are reduced.

Second, digital village construction can help guide the low-carbon development of agriculture from the demand side. On the one hand, the application of digital technology in the production and consumption of agricultural products has changed the current situation of weak competitiveness and high resource consumption of green agriculture. On the other hand, a breakthrough in market demand was achieved through the establishment of the agricultural product quality and safety traceability management information platform and the organic product certification traceability information system. The cultivation of green agricultural product network brands can drive the development of characteristic green agriculture and guide the low-carbon development of agriculture from the demand side.

Third, digital village construction has promoted financial support for agricultural green transformation. The main reason is that digital inclusive financial services have increased the application and promotion of green finance in rural areas, thereby promoting the progress and promotion of agricultural low-carbon technologies. At the same time, financial institutions use digital technology to control the risks of rural residents using low-carbon production technologies, reducing the cost of farmers adopting green production technologies.

Finally, digital village construction helps to form a good human living environment. Rural residents participate in environmental network supervision through the APP, thereby improving the monitoring ability of the water environment and water ecology in rural areas. The improvement of rural environmental disposal information management capabilities has improved the level of sewage and waste management. Therefore, digital village construction provides a basic guarantee for building a new development pattern of agricultural and rural green development.

### Mathematical model building

We assume that the introduction of technological advancement follows Harrod's neutral technology, and the general production form can be obtained as follows:

$$Y = F(K, DL) \tag{1}$$

Y is the total output, K is the total capital, L is the total labor force, and D is the digital village construction. Digital village construction can improve the allocation of production factors and increase labor productivity. Therefore, we regard the digital village construction level as the technical level of the economy. The production function is assumed to have constant returns to scale and diminishing marginal productivity. Eq (1) is a second-order classical

production function, continuously differentiable, and satisfies the following conditions:

$$F(\lambda K, \lambda L) = \lambda F(K, L), \lambda > 0 \tag{2}$$

$$F_1 > 0, F_{11} < 0, F_2 > 0, F_{22} < 0 \tag{3}$$

$$F(0, L) = F(K, 0) = 0, \lim_{x \to 0} F_i(x_1, x_2) = +\infty, \lim_{x \to \infty} F_i(x_1, x_2) = 0, i = 1, 2 \tag{4}$$

Based on the assumption of constant returns to scale, the production function can be rewritten in dense form as Eq (5):

$$\hat{y} = \frac{Y}{DL} = F\left(\frac{K}{DL}, 1\right) \equiv f\left(\hat{k}\right) \tag{5}$$

$\hat{k}$ is the capital stock of effective labor per capita. $\hat{y}$ is the output of effective labor per capita. Eq (5) gives the functional relationship between per capita effective capital and per capita effective output. Assuming that the labor force and digital village construction both grow at a constant rate:

$$\dot{L} = nL \tag{6}$$

$$\dot{D} = g_D D \tag{7}$$

We assume that total output is used for consumption and investment. The depreciation rate is δ. Then, we can obtain the following equation for capital change:

$$\dot{K} = sY - \delta K - C \tag{8}$$

s is the fixed savings rate. C is the cost of disposing of carbon dioxide. We assume that there are two ways to reduce carbon dioxide, one is to change the energy consumption structure, and the other is to increase the level of carbon sequestration. The cost formulas are shown in Eqs (9) and (10). Z is the firm's emissions, linked to the level of output. θ is the processing cost per unit of carbon emissions.

$$C = \theta z(Y) \tag{9}$$

$$\hat{C} \equiv \frac{C}{DL} = \frac{\theta z(Y)}{DL} = \theta z(\hat{y}) = \theta z\left(\hat{k}\right) \tag{10}$$

Substituting Eqs (9) and (10) into Eq (8), we can get Eq (11):

$$\frac{\dot{K}}{K} = \frac{SY}{K} - \delta - \frac{C}{K} = \frac{(sf(\hat{k}) - \hat{C})}{\hat{k} - \delta} \tag{11}$$

Therefore, the dynamic equation of effective capital per capita is written as Eq (12). Eq (13) is obtained by arranging Eq (12):

$$\frac{\dot{\hat{k}}}{\hat{k}} = \frac{\dot{K}}{K} - \frac{\dot{D}}{D} - \frac{\dot{L}}{L} = (sf(k) - \hat{C})\hat{k} - \delta - g_D - n \tag{12}$$

$$\dot{\hat{k}} = sf(\hat{k}) - \theta z(\hat{k}) - \hat{k}(\delta + g_D + n) \tag{13}$$

$$z\left(\hat{k}\right) = \frac{sf(\hat{k})}{\theta} - \frac{\hat{k}(\delta + g_D + n)}{\theta} \tag{14}$$

When the economy is in equilibrium, the growth rate of effective capital per capita is 0, which means $\dot{\hat{k}} = 0$. Therefore, the relationship between per capita effective carbon dioxide emissions and digital village construction is determined by Eq (14). In addition, the digital village growth rate and labor force growth rate are expressed as $g_D$ and $n$.

In Eq (14), the per capita effective carbon dioxide emissions are on the left. On the right-hand side of the equation, the first term means that the increase in output accumulates due to continued capital, thereby increasing carbon dioxide emissions. The second item means that with the continuous improvement of the digital village level, the carbon dioxide emissions in the production process are gradually reduced. The initial effective capital per capita is 0, which means that the firm does not produce and emits zero carbon dioxide. The inverted U-shaped relationship between CO2 emissions and digitization shows that as the digital village develops, $CO_2$ emissions first rise and then decrease, which is consistent with the assumptions of the environmental Kuznets curve.

In the initial stage of digital village construction, positive externalities such as technology spillovers and knowledge sharing are difficult to have a significant inhibitory effect on carbon emissions. This is reflected in the fact that the $CO_2$ emissions caused by the mass production of enterprises are greater than the $CO_2$ emissions reduced by digitalization, and the $CO_2$ emissions are faster than $CO_2$ processing. In addition, the factors and markets in rural areas are relatively scattered, the production and management costs of energy required by enterprises are relatively high, and economies of scale are not obvious. At the same time, rural residents have relatively low requirements for environmental quality, and environmental regulations are relatively loose, resulting in a continuous increase in carbon dioxide emissions.

With the continuous improvement of digital village construction, the $CO_2$ emission rate is lower than that of $CO_2$ treatment. The in-depth development of digital village construction has gradually revealed positive externalities such as technology spillover and knowledge sharing, which has brought about the optimization of the agricultural structure. Changes in the structure of the original energy-intensive and highly polluting heavy industries have significantly reduced carbon emissions.

**Hypothesis 1 (H1):** When other conditions remain unchanged, there is an "inverted U" curve relationship between digital rural construction and carbon emission.

## Path analysis of the impact of digital village construction on carbon emission

First, by adjusting the planting structure, the digital village construction can improve the carbon sink capacity of the land and reduce the carbon emission per unit of land. In the process of digitization of the agricultural chain, the structure of construction land has been continuously adjusted. At this time, the structure has shifted from high energy consumption and high

pollution emissions to the tertiary industry with low energy consumption and low pollution emissions. The optimization of this structure effectively reduces the carbon emissions per unit of construction land [16].

At the same time, with the emergence of modern technologies such as big data and cloud computing in rural areas, agricultural machinery has begun to integrate with modern information technology. It is manifested in the improvement of the intelligence level of traditional agricultural machinery and the improvement of the level of large-scale agricultural operations. In addition, food crops require fewer agricultural chemicals such as pesticides, fertilizers, and plastic films than commercial crops. Therefore, the increase in the grain planting area is beneficial to reducing the carbon emission per unit area and improving the land carbon sink capacity [17].

Secondly, digital village construction uses data as the input of production factors, which improves the efficiency of agricultural technology. Through the scale of agricultural production, promote the transformation of agriculture to green and low-carbon. The application of digital technology in rural resource development and production activities enables farmers to analyze production decisions through big data. This has greatly changed the traditional high-energy-consumption production methods and realized the intensification and refinement of the supply of agricultural resources.

This also avoids the problems of inefficient energy use and large input of pollutants in rural development. The scientific arrangement of agricultural seeding and fertilization is conducive to promoting changes in soil organic matter and improving soil carbon sequestration capacity [18]. At the same time, improvements in fertilization technology and farming methods have reduced the fertilizer input per unit of land, improved soil carbon sink capacity, and reduced plant carbon emissions [19].

Finally, digital village construction is conducive to the optimization of resource allocation and promotes the digital transformation of agriculture. The application of digital technology can effectively improve dynamic monitoring analysis. By promoting the key layout of carbon reduction areas such as livestock and poultry manure and crop straw recycling, the recycling of waste can be realized and the increase in carbon emissions caused by the random burning of waste can be reduced [20].

In addition, digital village construction can help alleviate information asymmetry. By reducing the energy consumption caused by intermediate links, efficient agricultural product information matching from "field " to "table" is realized. This greatly reduces the carbon emissions of transportation at the point of sale.

**Hypothesis 2 (H2):** The improvement of the planting structure helps to strengthen the carbon emission reduction effect of the digital village construction.

**Hypothesis 3 (H3):** The improvement of agricultural technology efficiency helps to strengthen the carbon reduction effect of digital village construction.

## Methods and data

### Measurement model settings

This paper aims to examine the impact of digital village construction on carbon emissions. The basic empirical model is shown in Eq (15):

$$\ln C_{it} = \alpha_0 + \alpha_1 sz_{it} + \alpha_2 sz_{it}^2 + \sum \beta_j cov_{it} + \mu_i + w_t + \varepsilon_{it} \tag{15}$$

The subscript $i$ represents the province. The subscript $t$ represents the year. $C_{it}$ is the carbon emission. $sz_{it}$ is the digital village construction. $cov_{it}$ is the control variable. $\mu_i$, $w_t$, and $\varepsilon_{it}$ are the individual fixed effects, time fixed effects, and random disturbance terms. Respectively. When $\alpha_1 > 0$ and $\alpha_2 < 0$, there is an "inverted U-shaped" relationship between digital village construction and carbon emission. When $\alpha_1 < 0$ and $\alpha_2 > 0$, there is a " U-shaped" relationship between digital village construction and carbon emission. Secondly, to test the mechanism of the effect of digital village construction on carbon emissions. This paper builds the following econometric model:

$$\ln c_{it} = \alpha_0' + \alpha_1' sz_{it} + \alpha_2' sz_{it}^2 + \gamma_1' M_{it} + \gamma_2' M_{it} \times sz_{it} + \sum \beta_j' cov_{it} + \mu_i' + w_t' + \varepsilon_{it}' \qquad (16)$$

$M_{it}$ are possible conduction paths. In this paper, crop planting structure and agricultural technical efficiency are used as proxy variables of M, respectively $\alpha_1'$. It is significantly greater than 0, $\alpha_2'$ significantly less than 0, and $\gamma_2'$ significantly negative, which means that M inhibits the promotion of carbon emissions in the early stage of digital village construction.

## Variable description

**CO2 emissions (CE).**  We discuss the carbon emission factors in rural areas based on the carbon emissions factors. Specifically, we divide rural carbon emission factors into living carbon sources, ecological carbon sources, and production carbon sources.

The living carbon sources mainly include the living conditions of rural residents and the consumption of durable goods needed in daily life, with rural housing, electricity, gasoline, and natural gas as the main emission factors. The ecological carbon sources include the improvement of transportation infrastructure and the improvement of air quality, etc., with roads, public buildings, productive buildings, and coal as the main emission factors. Production carbon sources include diesel fuel consumed by agricultural machinery and chemical fertilizers and pesticides in the agricultural production process, with diesel, pesticides, chemical fertilizers, and agricultural film as the main emission factors. The factors associated with carbon emissions are shown in Table 1.

**Digital village construction (DVC).**  In this paper, the index system is constructed based on the existing literature [13,21]. We plan to build an indicator system from four dimensions: digital village infrastructure construction, financial infrastructure construction, innovation capability, and service platform construction.

**Table 1. Rural carbon emission sources and parameters.**

| First-level indicator | Secondary indicators | Coefficient | Unit |
|---|---|---|---|
| **living carbon source** | rural housing | 0.0076 | tons/square meter/year |
| | electricity | 0.8560 | kg/degree |
| | gasoline | 2.9251 | ton/ton |
| | natural gas | 2.1622 | ton/ton |
| **ecological carbon source** | the way | 0.0047 | tons/square meter/year |
| | public building | 0.0076 | tons/square meter/year |
| | productive building | 0.0076 | tons/square meter/year |
| | coal | 1.9003 | ton/ton |
| **production carbon source** | diesel fuel | 3.0959 | ton/ton |
| | pesticide | 18.0800 | ton/ton |
| | fertilizer | 3.3000 | ton/ton |
| | Agricultural film | 18.9900 | ton/ton |

**Table 2. Index system of digital rural construction.**

| First-level indicator | Secondary indicators | Underlying indicator | Unit |
|---|---|---|---|
| digital village infrastructure construction | Broadband Internet Basics | rural broadband access users | million households |
| | Mobile Internet Basics | the average number of mobile phones per 100 people in rural households | department |
| financial infrastructure construction | Breadth of coverage | the coverage of digital financial inclusion | index |
| | Depth of use | depth of digital financial inclusion | index |
| innovation capability | Digital innovation element support | fiscal science and technology expenditure | billion |
| | Digital innovation output level | penetration of digital high-tech applications in listed companies | index |
| service platform construction | freight miles | rural delivery route length | kilometer |
| | E-commerce business development | number of Taobao Villages | individual |

In the digital village infrastructure construction, we consider the two dimensions of broadband Internet foundation and mobile Internet foundation and select rural broadband access households and the average number of rural households per 100 mobile phones as the underlying indicators. In the financial infrastructure construction, we characterize the coverage and depth of digital inclusive finance. In the innovation capability, we consider the support of digital innovation elements and the output level of digital innovation and select the local financial science and technology expenditure and the penetration of digital high-tech applications in listed companies as urban indicators. In the service platform construction, we used two indicators: the length of rural delivery routes(km) and the number of Taobao villages. The evaluation system of the digital village construction level is shown in Table 2.

**Control variables.** To control the impact of regional economic development and other factors on carbon emission, the following indicators are selected as control variables:

1. GDP per capita (*lg*) –According to the environmental Kuznets theory, the increase in carbon emissions is caused by economic growth by-products. With the improvement of the level of economic development, carbon emissions show a downward trend. Therefore, this paper takes the primary term *lg* and quadratic term *slg* of the logarithm of GDP per capita as control variables.

2. Energy consumption structure (*energy*) –This paper measures the proportion of coal consumption to total consumption. Coal consumption is the main source of fossil energy consumption. When the proportion of clean energy is larger, the carbon emission per unit of GDP is also lower.

3. Level of economic agglomeration (ln *jiju*) –We refer to the research of Ciccone and Hall [22], taking the labor force per unit of land area as a proxy variable of the level of economic agglomeration. Economic agglomeration can reduce carbon dioxide emissions through technological spillovers and economies of scale.

4. degree of opening to the outside world (*open*) –We measure by the ratio of FDI to GDP. The reason for considering this variable is mainly due to the FDI polluting paradise hypothesis.

**Endogenous mitigation and selection of instrumental variables.** This paper tries to control some key variables as much as possible. However, it is always difficult to prevent the occurrence of omitted variables which leads to possible estimation errors. For example, in areas with

a high level of digital village construction, the use of clean energy and rural governance capabilities are inherently high, so it is impossible to effectively identify whether digital village construction has an important impact on rural carbon emissions.

Secondly, digital village construction will affect rural carbon emissions, but rural carbon emissions will also limit the development of digital village construction, so there may be a reverse causality problem. Given this, we attempt to alleviate the endogeneity problem through the instrumental variable method. We consider the number of fixed telephones per 100 people and the volume of postal and telecommunications services per 100 people as instrumental variables for digital village construction.

Considerations for selecting instrumental variables are as follows.

First of all, the popularization of Internet technology is an important prerequisite for the construction of digital villages. One step forward, the application of Internet technology began with the popularization of landlines. Therefore, the construction of digital villages is closely related to the popularization of fixed-line telephones. That is to say, the areas with a high fixed-line penetration rate are very likely to be areas with better development of digital village construction. At the same time, the post office, as a directly related department of fixed-line installation, is also closely related to the construction of digital villages.

This paper uses the post and telecommunications information indicators of several years before as an instrumental variable. This processing assumes that the past post and telecommunications layout of a region will indirectly affect the current digital village construction by affecting the current Internet development, but it has little impact on today's carbon emissions. Therefore, using the number of fixed-line telephones in 1984 (Iv1) and the volume of postal and telecommunications services in 1984 (Iv2) as instrumental variables, the "correlation" and "exclusiveness" assumptions of effective instrumental variables are satisfied. Second, if the cross-sectional data is directly used as an instrumental variable, it will be difficult to measure due to the application of the fixed-effect model.

Therefore, this paper uses the intersection between the Internet penetration rate and the number of fixed telephones per 10,000 people, and the number of postal and telecommunications services per 100 people in 1984 as the instrumental variables for the current digital village construction.

**Handling of the indicator system.**   In this paper, the entropy method is used to assign values to the index system. To ensure the comparability of each index data, this paper carries out dimensionless normalization processing on the original data.

$$\text{Forward normalization}: \ X_{ij} = \frac{x_{ij} - x_{\min}}{x_{\max} - x_{\min}} + 0.01 \tag{17}$$

$$\text{Negative normalization}: \ X_{ij} = \frac{x_{\max} - x_{ij}}{x_{\max} - x_{\min}} + 0.01 \tag{18}$$

$X_{ij}$ is the value after the standardization of each indicator. $x_{ij}$ is the original value of the indicator. $x_{\min}$ and $x_{\max}$ represent the minimum and maximum values of the original data, respectively.

$$\text{The proportion of each indicator}: \ \beta_{ij} = \frac{X_{ij}}{\sum_{i=1}^{m} X_{ij}} \tag{19}$$

$$\text{Index entropy value}: \ e_{ij} = -\frac{1}{\ln m} \sum_{i=1}^{m} \beta_{ij} \ln \beta_{ij} \tag{20}$$

$$\text{Reverse entropy conversion}: \ p_i = 1 - e_i \tag{21}$$

$$\text{The indicator weights}: \ \omega_j = \frac{p_j}{\sum_{j=1}^{n} p_j} \tag{22}$$

The time factor is added on the basis of the traditional entropy method. The proportion of each index becomes $\beta_{pij} = \frac{X_{ij}}{\sum_{p=1}^{q} \sum_{i=1}^{m} X_{ij}}$, and the entropy value of the index changes as $e_{pij} = -\frac{1}{\ln m} \sum_{p=1}^{q} \sum_{i=1}^{m} \beta_{pij} \ln \beta_{pij}$. Based on this, the comprehensive index score of each province based on the time series can be obtained.

## Data sources

This paper takes 30 provinces in China from 2011 to 2020 as the research sample. The data mainly comes from the "China Rural Statistical Yearbook", "China Energy Statistical Yearbook" and "China Urban and Rural Construction Statistical Yearbook". We deflated and adjusted various currency volume indicators with 2011 as the base period to eliminate the impact of price factors. The descriptive statistics of each variable are shown in Table 3.

## Empirical results analysis

### Benchmark regression

Table 4 shows the regression results using random effects, fixed effects, and two-stage least squares, respectively. Hausman's test shows that it is more reasonable to use a fixed-effects model. Column (2) of Table 4 is the double fixed effect regression with the addition of control variables. The results show that the linear coefficient of digital rural construction is significantly positive, and the quadratic coefficient is significantly negative.

Columns (3)-(5) show the regression results using instrumental variables. After considering endogeneity, the elastic coefficients of the primary and quadratic terms of digital villages increase, and both are significant at the 1% level, indicating that the estimated coefficients of digital villages have a downward bias when endogeneity is not considered. In order to verify the validity of instrumental variables, the Kleibergen-Paaprk LM test, Kleibergen-Paaprk Wald F test, and Hansen J test were carried out. The choice of variables is appropriate.

So far, we can consider that hypothesis 1 holds. That is to say, there is a significant " inverted U-shaped" relationship between digital village construction and carbon emissions. In addition, the primary term of economic development level is significantly positive at the 5% level, and the quadratic term is significantly negative at the 10% level. It shows that with the growth of

**Table 3. Descriptive statistics of each variable.**

| Variable | N | Mean | Std. Dev | Min | Max |
|---|---|---|---|---|---|
| ln C | 300 | 4.6995 | 0.8918 | 1.7346 | 6.4951 |
| Sz | 300 | 0.1329 | 0.1149 | 0.0104 | 0.8180 |
| lg | 300 | 10.8406 | 0.4361 | 9.7058 | 12.0130 |
| energy | 300 | 0.9451 | 0.4441 | 0.0248 | 2.4609 |
| lnjiju | 300 | 0.2412 | 0.1183 | 0.1133 | 0.3867 |
| open | 300 | 0.0225 | 0.0182 | 0.0056 | 0.0594 |

**Table 4. Impact of digital economy development on rural carbon emission.**

| Variables | (1) | (2) | (3) | (4) | (5) |
|---|---|---|---|---|---|
| | RE | FE | The first stage | The first stage | Second stage |
| sz | 4.5996*** | 4.8409*** | | | 9.4004*** |
| | (1.3068) | (1.4213) | | | (1.3718) |
| sz2 | -3.6536*** | -3.8895** | | | -9.7546*** |
| | (1.4121) | (1.5254) | | | (1.8474) |
| lg | 8.2865** | 8.3425** | | | 5.9763*** |
| | (3.8741) | (3.9490) | | | (2.2997) |
| slg | -0.3598** | -0.3651* | | | -0.2788*** |
| | (0.1772) | (0.1811) | | | (0.1045) |
| energy | -0.2693 | -0.3086 | | | -0.3275** |
| | (0.3166) | (0.3758) | | | (0.1560) |
| lnjiju | 0.0021 | 0.0026 | | | 0.0035 |
| | (0.0024) | (0.0024) | | | (0.0071) |
| open | -0.1091 | -0.2032 | | | -0.9915 |
| | (3.5575) | (3.5882) | | | (1.2093) |
| IV1 | | | 0.0075*** | 0.0062*** | |
| | | | (0.0007) | (0.0009) | |
| IV2 | | | -0.0756** | -0.1640*** | |
| | | | (0.0320) | (0.0385) | |
| Constant | -43.0920** | -43.1185* | | | -28.1953** |
| | (21.1972) | (21.5353) | | | (12.3968) |
| Observations | 300 | 300 | 300 | 300 | 300 |
| R-squared | 0.7026 | 0.7029 | | | 0.9429 |
| Province FE | | YES | YES | YES | YES |
| Year FE | | YES | YES | YES | YES |
| rk Lm | | | | 21.337*** | |
| rk F | | | | 14.594 | |
| Hansen J | | | | 0.000 | |

Note

\*\*\*, \*\*, and \* represent significance levels of 1%, 5%, and 10% respectively. The values in brackets are robust standard errors.

the economic level, carbon emissions will first show a trend of increasing first and then decreasing. This is consistent with the environmental Kuznets hypothesis.

The overall conclusion shows that the hypothesis about Kuznets also occurs in the context of digital economic activity. This is similar to the conclusion of Meng and Zhao [6]. Next, this paper will further discuss the internal mechanism of the impact of digital village construction on carbon emissions.

## Fractional test

Table 5 reports the impact of digital village construction on sub-dimension carbon emissions. The results in columns (1) and (2) of Table 5 show that there is an obvious "inverted U-shaped" relationship between digital village construction and living carbon sources and ecological carbon sources. This shows that the construction of digital villages will increase living carbon emissions and ecological carbon emissions in rural areas in the short term. But after a certain period, the opposite happens. The possible reason is that the construction of digital villages has increased the demand for infrastructure such as electricity and roads in rural areas, resulting

Table 5. The impact of digital villages on carbon emissions in various dimensions.

| Variables | (1) | (2) | (3) |
|---|---|---|---|
| | living carbon source | ecological carbon source | production carbon source |
| sz | 9.2158*** | 0.0560** | -0.0943 |
| | (1.3445) | (0.0261) | (0.0578) |
| sz2 | -9.5150*** | -0.0813** | 0.0141 |
| | (1.8139) | (0.0339) | (0.0715) |
| Constant | -28.5471** | 2.7686*** | 0.3416 |
| | (12.4124) | (0.3900) | (0.3100) |
| Control variables | YES | YES | YES |
| Province FE | YES | YES | YES |
| Year FE | YES | YES | YES |
| Observations | 300 | 300 | 300 |
| R-squared | 0.943 | 0.974 | 0.989 |

in a rapid rise in carbon emissions from electricity, roads, and public buildings. This is similar to the conclusion of Li et al. [23]. Therefore, in the initial stage, digital village construction will lead to the increase of living carbon sources and ecological carbon sources in rural areas.

With the development of digital village construction, residents' digital literacy and environmental awareness are gradually enhanced. The use of coal and gasoline has gradually declined. Clean energy is gradually replacing coal, and traditional vehicles are gradually being replaced by new energy vehicles. These have greatly eased the pressure of carbon emission reduction in rural areas, thereby reducing domestic carbon emissions and ecological carbon emissions. Therefore, there is an "inverted U-shaped" relationship between digital village construction and domestic carbon emissions, and ecological carbon emissions.

The results in column (3) of Table 5 show that the impact of digital village construction on production carbon emissions is not significant. The possible reason is that for areas with high population density and weak infrastructure construction, the carbon reduction effect of digital technology in the production field is limited, and the scale effect is not obvious. Therefore, the effect of digital village construction on production carbon emissions is not obvious. This is similar to the conclusion of Balsalobre-Lorente et al. [24].

## Conduction mechanism test

**Conduction path of crop planting structure.** In this paper, the ratio of food crop cultivation area and agricultural land area is used as the measurement variable of the planting structure. The results in column (1) of Table 6 show that the cross-product of digital rural construction and planting structure is significant at the 10% level, indicating that the larger proportion of grain crop area restrains the increase in carbon emissions in the early stage of digital rural construction.

The results in column (2) of Table 6 show that the optimization of crop planting structure promotes the earlier arrival of the "inverted U-shaped" inflection point between digital village construction and rural carbon emission. Therefore, Hypothesis 2 is established.

**Conduction path of agricultural technical efficiency.** We use DEAP2.1 software to measure agricultural technical efficiency. The capital input selects three indicators: the effective irrigation area, the total power of agricultural machinery, and the number of agricultural chemical fertilizers. The labor input selects two indicators such as the area of arable land and the number of laborers. The output indicator is the gross agricultural product.

**Table 6. Conduction mechanism test.**

| Variables | (1) | (2) | (3) | (4) |
|---|---|---|---|---|
| | lnc | lnc | lnc | lnc |
| sz | 7.7547*** | 7.8982*** | 7.7764*** | 7.8017*** |
| | (0.5623) | (0.5586) | (0.5394) | (0.5362) |
| sz2 | -7.1093*** | -7.3752*** | -7.0979*** | -7.1793*** |
| | (0.7900) | (0.7601) | (0.7854) | (0.7741) |
| zz | 0.0414 | 0.0811 | | |
| | (0.4822) | (0.4817) | | |
| sz*zz | -13.1556* | | | |
| | (7.6634) | | | |
| sz2*zz | | -28.6005** | | |
| | | (13.0680) | | |
| js | | | 0.8670*** | 0.9002*** |
| | | | (0.2978) | (0.3070) |
| sz*js | | | -7.1947** | |
| | | | (3.3494) | |
| sz2*js | | | | -11.8637** |
| | | | | (4.8435) |
| Constant | -0.0297 | -0.0166 | -0.0000 | -0.0000 |
| | (0.0459) | (0.0419) | (0.0383) | (0.0383) |
| Control variables | YES | YES | YES | YES |
| Province FE | YES | YES | YES | YES |
| Year FE | YES | YES | YES | YES |
| Observations | 300 | 300 | 300 | 300 |
| R-squared | 0.647 | 0.649 | 0.665 | 0.665 |

The results in columns (3) and (4) of Table 6 show the impact of the improvement of agricultural technical efficiency on the carbon emission reduction effect of digital villages. The results in column (3) of Table 6 show that the interaction term between digital villages and agricultural technical efficiency is significant at the level of 5%, indicating that the improvement of agricultural technical efficiency has shortened the range of positive effects of digital village construction on carbon emissions.

The results in column (4) of Table 6 show that the interaction term between the quadratic term of digital rural construction and agricultural technical efficiency is significant at the 5% level. Moreover, the regression results are consistent with the benchmark regression, indicating that the improvement of agricultural technical efficiency has strengthened the carbon emission reduction effect of digital village construction. Therefore, Hypothesis 3 is established.

## Heterogeneity analysis

**Analysis based on the degree of economic development.** From a factual point of view, the level of economic development is an important factor affecting carbon emissions. In regions with relatively high levels of economic development, the carbon emission reduction mechanisms are relatively more complete, and the driving force for reducing pollution emissions through innovative emission reduction technologies is stronger. And a higher level of economic development is often accompanied by a higher level of digital supervision. Therefore, we will discuss the differential impact of digital village construction on carbon emission reduction under different economic development levels.

Table 7. Heterogeneity analysis.

| Variables | GDP per capita | | | environmental regulation | | |
|---|---|---|---|---|---|---|
| | (1) | (2) | (3) | (4) | (5) | (6) |
| | High-level | Middle-level | Low-level | High-level | Middle-level | Low-level |
| sz | 11.8224*** | 11.4682*** | 8.0387** | 10.1851 | 6.8592** | 5.3152*** |
| | (2.1400) | (2.9985) | (3.1432) | (5.6650) | (2.2058) | (1.4288) |
| sz2 | -11.9876*** | -13.3287* | -9.4254 | -15.5733 | -17.0751* | -4.2475** |
| | (2.3127) | (6.9082) | (12.0191) | (14.9111) | (8.1548) | (1.4590) |
| Constant | 101.6691** | 56.6140** | -97.5503*** | -88.3703 | -13.2852 | -9.0598 |
| | (49.9649) | (26.3027) | (37.0742) | (67.2601) | (14.2350) | (20.4857) |
| Control variables | YES | YES | YES | YES | YES | YES |
| Province FE | YES | YES | YES | YES | YES | YES |
| Year FE | YES | YES | YES | YES | YES | YES |
| Observations | 100 | 100 | 100 | 100 | 100 | 100 |
| R-squared | 0.810 | 0.975 | 0.956 | 0.654 | 0.881 | 0.849 |

The regions with the median value in the top 1/3 of the sample period were defined as high levels, the cities with per capita GDP in the lower 1/3 were defined as low levels, and the middle regions were defined as medium regions, and sub-sample regression was performed.

The sub-sample regression results are shown in Table 7. The estimation results in columns (1) and (2) in Table 7 show that in the regions with high economic development levels, the coefficients of the first and quadratic terms of digital village construction are 11.8224 and -11.9876 respectively, which are significant at the 1% level is positive. In areas with a moderate level of economic development, the primary and quadratic coefficients of digital village construction are 11.4682 and 13.3287, which are significant at the level of 1% and 10%, respectively.

The results in column (3) of Table 7 show that the coefficient of the first order of digital village construction is 8.0387 at the 5% level, but the coefficient of the second order is not significant. The results show that the impact of digital village construction on carbon emission reduction in areas with high economic development levels is stronger than that in areas with low economic development levels. This is similar to the conclusion of Salahuddin et al. (2016).

**Analysis based on the intensity of environmental regulation.** The effectiveness of China's carbon emission reduction system is affected by environmental regulation. Strong environmental regulations provide a certain guarantee for the effective operation of emission reduction work. We take environmental protection tax as a proxy variable for environmental regulation and use resource tax, urban construction tax, land tax, consumption tax, etc. as the main object of environmental protection tax. The estimated results are shown in columns (4) to (6) of Table 7.

Column (4) in Table 7 shows that in areas with a relatively high degree of environmental regulation, the carbon emission reduction effect of digital village construction is not obvious. The possible explanation is that generally speaking, areas with strong environmental regulations, often face greater environmental pressures. This will also have a greater incentive effect on energy conservation and emission reduction for digital village construction. But the reality is that the effect of environmental regulation depends not only on the intensity of environmental regulation but also on the form of environmental regulation. When the form of regulation is unreasonable, even if the government and society pay more attention to the carbon emission problem in agriculture and rural areas, the desired effect may not be achieved, thus reducing the significance of the "U-shaped" curve trend.

The results in columns (5) and (6) of Table 7 show that in areas with a moderate degree of environmental regulation, the primary and quadratic coefficients of digital village construction are 6.8592 and -17.0715, which are significant at the level of 5% and 1%, respectively. In areas with low environmental regulation, the primary and quadratic coefficients of digital village construction are 5.3152 and -4.2475, which are significant at the level of 1% and 5%, respectively. Through comparison, it is found that, compared with areas with low environmental regulation, the impact of digital village construction on carbon emission reduction in areas with medium environmental regulation is more obvious. This is similar to the findings of Hashmi and Alam [25].

### Further robustness check

Excluding the municipalities takes into account the obvious political, economic, and cultural advantages of the municipalities directly under the Central Government. To eliminate the influence of this institutional factor on the regression results, we remove the sample of municipalities and use the remaining samples for regression. The regression results are shown in column (1) of Table 8. There is still an "inverted U-shaped" relationship between digital village construction and rural carbon emission.

In the previous analysis, we used the total amount of rural carbon emission as the explained variable. To eliminate the difference in results caused by population, we further used the rural per capita carbon emission as the explained variable to estimate the parameters.

The results in column (2) of Table 8 show that the digital village construction and rural per capita carbon emission still show an "inverted U-shaped" relationship. In addition, we also use the unit GDP carbon emission as the explained variable for parameter estimation. The results in Table 8(3) show that the conclusion of the benchmark study in this paper is still valid.

### Conclusion and discussion

Over the past 40 years of reform and opening up, China has made achievements in economic and social development that have attracted worldwide attention. This is accompanied by China's growing energy demand and relatively serious environmental pollution, which also greatly restricts the high-quality development of China's economy and society and even other countries. China now ranks second in the world in terms of GDP, but first in pollutant emissions

**Table 8. Robustness test.**

| Variables | (1) | (2) | (3) |
|---|---|---|---|
| | Exclude municipalities | Carbon emissions per capita | Carbon emissions per unit of GDP |
| sz | 10.1162*** | 2.9658*** | 0.0170*** |
| | (1.7359) | (1.0027) | (0.0058) |
| sz2 | -9.8937*** | -4.1107*** | -0.0138** |
| | (2.3229) | (1.3283) | (0.0055) |
| Constant | -70.7781*** | 36.7175*** | -0.0093 |
| | (18.0521) | (6.3771) | (0.0466) |
| Control variables | YES | YES | YES |
| Observations | 260 | 300 | 300 |
| R-squared | 0.949 | 0.860 | 0.351 |
| Number of ids | 26 | 30 | 30 |
| Province FE | YES | YES | YES |
| Year FE | YES | YES | YES |

(CO2 SO2, PM2.5, nitrogen oxides, etc.) and primary energy consumption, which forces China to shift from traditional development to green development, and the three key factors of green development are economic development, resource conservation, and environmental protection. Although a large number of studies have been conducted on energy conservation and emission reduction in China in the existing literature to seek better environmental regulatory measures, they have paid little attention to the situation in agriculture and rural areas and conducted relevant research.

In this paper, 30 provinces and cities in China are selected as observation samples, and the impact of digital village construction on rural carbon emissions is empirically tested by alleviating endogeneity through instrumental variables. The mechanism of action and implementation effect were analyzed by mediating effect model and heterogeneity test. The key results are as follows:

Firstly, we find that there is a significant "inverted U" curve between digital village construction and rural carbon emissions. This result shows that when the level of digital villages is low, the increase in digital villages will increase rural carbon emissions. However, when the level of digital village exceeds the critical value, the improvement of the digital village level will reduce rural carbon emissions, proving that the relationship between digital village construction and rural carbon emissions follows the environmental Kuznets curve hypothesis. Therefore, local governments should step up the formulation of local digital village construction and development plans, lead the construction and development of digital villages with high-standard planning, accelerate the construction of digital village information infrastructure, orderly promote the construction of 5G and gigabit Internet in rural areas, and promote the inflection point of the growth curve as soon as possible. Accelerate the digital and intelligent transformation of infrastructure such as water conservancy, highways, and electricity in rural areas, provide a foundation for application scenarios such as smart agricultural production, rural e-commerce, and digital life, and provide basic support for the realization of green, accurate, and smart agriculture.

In addition, there is heterogeneity in the impact effect of digital village construction on different carbon sources, and the emission reduction efforts of different carbon sources are also different. Therefore, local governments should formulate reasonable emission reduction targets, formulate effective emission reduction targets based on the actual conditions in rural areas, and gradually realize the coordinated symbiosis of rural economic development and ecological protection.

Secondly, we observe that regional environmental regulation will have an impact on the carbon emission reduction effect of digital village construction, and the weaker the carbon emission reduction effect of the digital village in areas with high environmental regulation, which contradicts the goal of the environmental regulation strategy. This result is not surprising, because the effectiveness of environmental regulation depends not only on the intensity of environmental regulation but also on the form of environmental regulation. When the regulatory form is unreasonable, even if the government and society pay more attention to the overall problem of agricultural and rural carbon emissions, it may not achieve the desired effect, thereby reducing the significance of the "U" curve trend. Therefore, local governments should pay attention to the form of reasonable environmental regulation while promoting the construction of digital villages. In the process of promoting the construction of digital villages, the relationship between agricultural production and environmental protection should be coordinated to achieve green and low-carbon development of agriculture.

Crop planting results and agricultural technology efficiency are important ways for digital village construction to affect carbon emissions. Digital villages empower the entire agricultural industry chain through digitalization, improve the level of agricultural mechanization,

rationalize the planting structure and improve agricultural technology efficiency, improve soil carbon sink capacity and reduce pollutant emissions. Local governments should actively promote conservation tillage, straw return, organic fertilizer application, artificial grass planting, and other measures, strengthen the construction of high-standard farmland, increase soil organic matter content, and improve greenhouse gas absorption and fixation capacity. At the same time, it is necessary to promote advanced and applicable low-carbon energy-saving agricultural machinery and equipment to reduce fossil energy consumption and carbon dioxide emissions; In the subsidy catalog for the purchase of agricultural machinery, increase the performance requirements for energy conservation of agricultural machinery, and support the research and development and promotion of energy-saving agricultural machinery.

However, regardless of the positive results, there are still some limitations. First, due to China's vast territory, there are obvious differences in resource endowments and energy consumption structures in different regions, which may distort the research conclusions. Therefore, it is necessary to conduct case studies for specific regions to enrich the empirical evidence on the impact of digital village construction on carbon emission reduction. Second, compared with the spread of digital information technology in rural areas, we should also pay attention to how rural residents' digital literacy can help achieve the goal of carbon neutrality. This is also an important direction for the next step of research.

## Supporting information

**S1 Dataset.**
(XLSX)

## Author Contributions

**Data curation:** Jiayin Tan.

**Formal analysis:** Yirui Hou.

**Funding acquisition:** Aimin Hao.

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
