## [Decision Letter · Decision Letter 0]

24 Oct 2022

PONE-D-22-26840How does Digital Village Construction Influences Carbon Emission? The Case of ChinaPLOS ONE

Dear Dr. Yirui Hou,

Thank you for submitting your manuscript to PLOS ONE. After careful consideration, we feel that it has merit but does not fully meet PLOS ONE’s publication criteria as it currently stands. Therefore, we invite you to submit a revised version of the manuscript that addresses the points raised during the review process.

We look forward to receiving your revised manuscript.

Kind regards,

Carlos Alberto Zúniga-González, Ph.D

Academic Editor

PLOS ONE

Additional Editor Comments:

Dear author, the observation is led to make some change in your manuscript. Is very important to follow the observations of second reviewer. I suggest that you highlight the objective of your research and try to focus your conclusions on and about your objective.

Reviewers' comments:

Reviewer's Responses to Questions

**Comments to the Author**

1. Is the manuscript technically sound, and do the data support the conclusions?

Reviewer #1: Yes

Reviewer #2: No

2. Has the statistical analysis been performed appropriately and rigorously? 

Reviewer #1: Yes

Reviewer #2: N/A

3. Have the authors made all data underlying the findings in their manuscript fully available?

Reviewer #1: Yes

Reviewer #2: Yes

4. Is the manuscript presented in an intelligible fashion and written in standard English?

Reviewer #1: Yes

Reviewer #2: Yes

5. Review Comments to the Author

Reviewer #1: In my opinion, the authors presented sufficient evidence that supports the verification of the hypotheses raised in the investigation.Above all, they they checked that there is a significant "inverted U-shaped" relationship between digital village construction and rural carbon emission. In addition, they verified the positive impact of the the construction of digital villages.

Reviewer #2: The author must follow the writing guidelines.

the paper did nos show objectives or goals

Too much general information is quoted

The discussion and conclusion parts should be separated for analysis.

The conclusion are very extensive and part of the that information is cited in methodology

It is suggested that it be written impersonally. Make excessive use of "We"

Likewise, it is exhorted not to write such long paragraphs, or to put two or three paragraphs together. It is difficult to read

Finally, this is a revision work, It may have to be published as a scientific note after the author makes the observation indicated in the text

6. PLOS authors have the option to publish the peer review history of their article (what does this mean?). If published, this will include your full peer review and any attached files.

Reviewer #1: **Yes: **Napoleon Vicente Blanco Orozco

Reviewer #2: No

---

## [Author Response · Author response to Decision Letter 0]

11 Nov 2022

Dear Editors and Reviewers:

Thank you for your letter and for the reviewer’s comments concerning our manuscript entitled “How does Digital Village Construction Influences Carbon Emission? The Case of China”. Those comments are all valuable and very helpful for revising and improving our papers, as well as the important guiding significance to our researches. We have studied comments carefully and have made correction which we hope meet with approval. Revised portion are marked in red in the paper. The main corrections in the paper and the responds to the reviewer’s comments are as flowing:

Responds to the reviewer’s comments:

Reviewer 1:

1). Response to comment: In my opinion, the authors presented sufficient evidence that supports the verification of the hypotheses raised in the investigation.Above all, they they checked that there is a significant "inverted U-shaped" relationship between digital village construction and rural carbon emission. In addition, they verified the positive impact of the the construction of digital villages.

Response: Thank you for your comments. We will further polish the manuscript.

Reviewer 2:

1). Response to comment:The author must follow the writing guidelines.

Response: Thank you for your comments. Considering your suggestion, the format of the paper has been revised and the content has been polished

2). Response to comment: The paper did nos show objectives or goals.

Response: Thank you for your comments. Considering your suggestion , we have added some content according to your comments. Specifically, “Therefore, it is undoubtedly of great significance to explore an effective path for China to achieve the goal of carbon neutrality.” (Lines 30-31)

“At present, what needs to be explored is whether China's digital village construction can affect the carbon neutrality and carbon peaking strategy, and if so, what are the characteristics of the economic mechanism and effect behind it? In China's specific context, does the relationship between digital village construction and rural carbon emissions also follow the environmental Kuznets curve (EKC) hypothesis? The exploration of these issues is not only related to the realization of China's "dual carbon" goal strategy but also has important reference significance for all countries in the world to achieve the goal of carbon neutrality.” (Lines 42-48)

3). Response to comment: Too much general information is quoted.

Response: Thank you for your comments. We are very sorry for our negligence of the brevity of the sentences in the paper. Superfluous general information have been removed from the article to make the content of the article more concise.

4). Response to comment: The discussion and conclusion parts should be separated for analysis.The conclusion are very extensive and part of the that information is cited in methodology.

Response: Thank you for your comments. Considering your suggestion, we have re-written this part according to your comments. Specifically, “Over the past 40 years of reform and opening up, China has made achievements in economic and social development that have attracted worldwide attention. This is accompanied by China's growing energy demand and relatively serious environmental pollution, which also greatly restricts the high-quality development of China's economy and society and even other countries. China now ranks second in the world in terms of GDP, but first in pollutant emissions (CO2 SO2, PM2.5, nitrogen oxides, etc.) and primary energy consumption, which forces China to shift from traditional development to green development, and the three key factors of green development are economic development, resource conservation, and environmental protection. Although a large number of studies have been conducted on energy conservation and emission reduction in China in the existing literature to seek better environmental regulatory measures, they have paid little attention to the situation in agriculture and rural areas and conducted relevant research.

In this paper, 30 provinces and cities in China are selected as observation samples, and the impact of digital village construction on rural carbon emissions is empirically tested by alleviating endogeneity through instrumental variables. The mechanism of action and implementation effect were analyzed by mediating effect model and heterogeneity test. The key results are as follows:

Firstly, we find that there is a significant "inverted U" curve between digital village construction and rural carbon emissions. This result shows that when the level of digital villages is low, the increase in digital villages will increase rural carbon emissions. However, when the level of digital village exceeds the critical value, the improvement of the digital village level will reduce rural carbon emissions, proving that the relationship between digital village construction and rural carbon emissions follows the environmental Kuznets curve hypothesis. Therefore, local governments should step up the formulation of local digital village construction and development plans, lead the construction and development of digital villages with high-standard planning, accelerate the construction of digital village information infrastructure, orderly promote the construction of 5G and gigabit Internet in rural areas, and promote the inflection point of the growth curve as soon as possible. Accelerate the digital and intelligent transformation of infrastructure such as water conservancy, highways, and electricity in rural areas, provide a foundation for application scenarios such as smart agricultural production, rural e-commerce, and digital life, and provide basic support for the realization of green, accurate, and smart agriculture.

In addition, there is heterogeneity in the impact effect of digital village construction on different carbon sources, and the emission reduction efforts of different carbon sources are also different. Therefore, local governments should formulate reasonable emission reduction targets, formulate effective emission reduction targets based on the actual conditions in rural areas, and gradually realize the coordinated symbiosis of rural economic development and ecological protection.

Secondly, we observe that regional environmental regulation will have an impact on the carbon emission reduction effect of digital village construction, and the weaker the carbon emission reduction effect of the digital village in areas with high environmental regulation, which contradicts the goal of the environmental regulation strategy. This result is not surprising, because the effectiveness of environmental regulation depends not only on the intensity of environmental regulation but also on the form of environmental regulation. When the regulatory form is unreasonable, even if the government and society pay more attention to the overall problem of agricultural and rural carbon emissions, it may not achieve the desired effect, thereby reducing the significance of the "U" curve trend. Therefore, local governments should pay attention to the form of reasonable environmental regulation while promoting the construction of digital villages. In the process of promoting the construction of digital villages, the relationship between agricultural production and environmental protection should be coordinated to achieve green and low-carbon development of agriculture.

Crop planting results and agricultural technology efficiency are important ways for digital village construction to affect carbon emissions. Digital villages empower the entire agricultural industry chain through digitalization, improve the level of agricultural mechanization, rationalize the planting structure and improve agricultural technology efficiency, improve soil carbon sink capacity and reduce pollutant emissions. Local governments should actively promote conservation tillage, straw return, organic fertilizer application, artificial grass planting, and other measures, strengthen the construction of high-standard farmland, increase soil organic matter content, and improve greenhouse gas absorption and fixation capacity. At the same time, it is necessary to promote advanced and applicable low-carbon energy-saving agricultural machinery and equipment to reduce fossil energy consumption and carbon dioxide emissions; In the subsidy catalog for the purchase of agricultural machinery, increase the performance requirements for energy conservation of agricultural machinery, and support the research and development and promotion of energy-saving agricultural machinery.

However, regardless of the positive results, there are still some limitations. First, due to China's vast territory, there are obvious differences in resource endowments and energy consumption structures in different regions, which may distort the research conclusions. Therefore, it is necessary to conduct case studies for specific regions to enrich the empirical evidence on the impact of digital village construction on carbon emission reduction. Second, compared with the spread of digital information technology in rural areas, we should also pay attention to how rural residents' digital literacy can help achieve the goal of carbon neutrality. This is also an important direction for the next step of research.” (Lines 549-617)

5). Response to comment: It is suggested that it be written impersonally. Make excessive use of "We".Likewise, it is exhorted not to write such long paragraphs, or to put two or three paragraphs together. It is difficult to read

Response: Thank you for your comments. We have revised the long paragraphs in the article to make it easier for readers to read. At the same time, The content has been edited to reduce non-subjective descriptions.

6). Response to comment: Finally, this is a revision work, It may have to be published as a scientific note after the author makes the observation indicated in the text.

Response: Thank you for your comments. We will take your comments seriously.

---

## [Decision Letter · Decision Letter 1]

18 Nov 2022

How does Digital Village Construction Influences Carbon Emission? The Case of China

PONE-D-22-26840R1

Dear Dr. Yirui Hou,

We’re pleased to inform you that your manuscript has been judged scientifically suitable for publication and will be formally accepted for publication once it meets all outstanding technical requirements.

Kind regards,

Carlos Alberto Zúniga-González, Ph.D

Academic Editor

PLOS ONE

Additional Editor Comments (optional):

Dear I am checking that you have been added all the observations' reviewers. Congratulations, !!!!!!!

Reviewers' comments:

Reviewer's Responses to Questions

**Comments to the Author**

1. If the authors have adequately addressed your comments raised in a previous round of review and you feel that this manuscript is now acceptable for publication, you may indicate that here to bypass the “Comments to the Author” section, enter your conflict of interest statement in the “Confidential to Editor” section, and submit your "Accept" recommendation.

Reviewer #2: All comments have been addressed

2. Is the manuscript technically sound, and do the data support the conclusions?

Reviewer #2: Yes

3. Has the statistical analysis been performed appropriately and rigorously? 

Reviewer #2: Yes

4. Have the authors made all data underlying the findings in their manuscript fully available?

Reviewer #2: Yes

5. Is the manuscript presented in an intelligible fashion and written in standard English?

Reviewer #2: Yes

6. Review Comments to the Author

Reviewer #2: Minimum details have been indicated in the document. The author is suggested to review the use of et al. when it goes inside and outside the parentheses

7. PLOS authors have the option to publish the peer review history of their article (what does this mean?). If published, this will include your full peer review and any attached files.

Reviewer #2: No

---

## [Editor Report · Acceptance letter]

1 Dec 2022

PONE-D-22-26840R1 

How does Digital Village Construction Influences Carbon Emission? The Case of China 

Dear Dr. Hou:

I'm pleased to inform you that your manuscript has been deemed suitable for publication in PLOS ONE. Congratulations! Your manuscript is now with our production department. 

Kind regards, 

on behalf of

Dr. Prof. Carlos Alberto Zúniga-González 

Academic Editor

PLOS ONE